# The rise of open data practices among bioscientists at the University of Edinburgh

Haya Deeb[1], Suzanna Creasey[1], Diego Lucini de Ugarte[1], George Strevens[1],
Trisha Usman[1], Hwee Yun Wong[1], Megan A. M. Kutzer[1,2], Emma Wilson[3],
Tomasz Zieliński[1], Andrew J. Millar[1]*

1 Centre for Engineering Biology and School of Biological Sciences, University of Edinburgh, Edinburgh, United Kingdom, 2 Institute of Ecology and Evolution and School of Biological Sciences, University of Edinburgh, Edinburgh, United Kingdom, 3 Centre for Clinical Brain Sciences, University of Edinburgh, Edinburgh, United Kingdom

* andrew.millar@ed.ac.uk

## Abstract

Open science promotes the accessibility of scientific research and data, emphasising transparency, reproducibility, and collaboration. This study assesses the Openness and FAIR (Findable, Accessible, Interoperable, and Reusable) aspects of data-sharing practices within the biosciences at the University of Edinburgh from 2014 to 2023. We analysed 555 research papers across biotechnology, regenerative medicine, infectious diseases, and non-communicable diseases. Our scoring system evaluated data completeness, reusability, accessibility, and licensing, finding a progressive shift towards better data-sharing practices. The fraction of publications that share all relevant data increased significantly, from 7% in 2014 to 45% in 2023. Data involving genomic sequences were shared more frequently than image data or data on human subjects or samples. The presence of data availability statement (DAS) or preprint sharing correlated with more and better data sharing, particularly in terms of completeness. We discuss local and systemic factors underlying the current and future Open data sharing. Evaluating the automated ODDPub (Open Data Detection in Publications) tool on this manually-scored dataset demonstrated high specificity in identifying cases where no data was shared. ODDPub sensitivity improved with better documentation in the DAS. This positive trend highlights improvements in data-sharing, advocating for continued advances and addressing challenges with data types and documentation.

## Introduction

Open science is the movement aimed at making scientific research, data, and dissemination more universally accessible. It encompasses a range of practices that promote transparency, reproducibility and collaboration, including open access to

**Data availability statement:** All data utilised in this study, along with the detailed code, are available in the Zenodo repository dedicated to this paper (https://doi.org/10.5281/zenodo.14169649 ) and should be cited as (APA Style): "Deeb, H., Creasey, S., de Ugarte, D. L., Strevens, G., Usman, T., Yun Wong, H., Kutzer, M. A. M., Wilson, E., Zieliński, T., & Millar, A. J. (2024). A Decade of Progress: Open Data Practices in Bioscience at the University of Edinburgh (1.0) [Data set]. Zenodo. https://doi.org/10.5281/zenodo.14169649. " The main repository of the project with all the previous data and output can be found on Github: BioRDM/InsightsOfOpenPracticesInBiosciences [Internet]. BioRDM; 2024 [cited 2025 Jan 7]. Available from: https://github.com/BioRDM/InsightsOfOpenPracticesInBiosciences. The detailed scoring protocol can be retrieved and adapted from the protocols.io repository: https://dx.doi.org/10.17504/protocols.io.kx-ygxyxmdl8j/v2. The preprint of the article can be found on BioRxiv.

**Funding:** This work was supported by UK Research & Innovation - Medical Research Council (MR/X009726/1 to AJM and others), the Wellcome Trust (Mental Health Award AMBIENT-BD 26944/Z/23/Z to AJM and others), the School of Biological Sciences, University of Edinburgh (BioRDM team funds to AJM) and by a PhD studentship from the Simons Initiative for the Developing Brain (SFARI #529085 to EW).

**Competing interests:** The authors have declared that no competing interests exist.

publication, open peer review, and open data [1]. Open data, in particular, is a critical component of open science, as it ensures that the data underlying research findings are freely available for scrutiny, validation, and reuse by others in the scientific community [2]. By making data openly available, researchers can enhance the reproducibility of their findings, facilitate discoveries through data reuse, and promote greater transparency in scientific research [3,4].

The concept of FAIR data – emphasising data that is 'Findable', 'Accessible', 'Interoperable', and 'Reusable' – [5,6] was introduced in 2016 to promote the transparency and integrity of shared research data, even if those data are not fully Open. Simply being 'open' does not guarantee that data are well-organised, thoroughly documented, or easily re-usable [7], so the FAIR principles are now recognised as widely applicable and provide one focus for an ongoing shift in policies regarding data sharing. Major funding agencies such as the National Institutes of Health (NIH) and the European Commission have implemented policies that require data sharing as a condition of funding [8,9]. Journals such as PLOS ONE and Nature have mandated Data Availability Statements for all research articles, requiring authors to clearly specify how and where their data can be accessed [10,11].

Despite the benefits and policy support, several challenges persist in data sharing. Researchers may be reluctant to share data due to concerns about data misuse, loss of competitive advantage, or the significant time and effort required to prepare data for sharing [12]. The lack of standardised formats and repositories can complicate data-sharing efforts, while intellectual property rights and considerations surrounding human data disclosure pose consistent challenges [13–17]. A very high prevalence of data sharing has been reported in specific cases, for example over 50% of narrowly-sampled publications in 2010−2019 shared sequence of a single gene (HIV-1 *pol*) that is monitored for viral drug resistance [18]. A study of experimental zoology reported data sharing prevalence of 19% [19], among the higher rates reported, compared to 8% in a large meta-analysis of biomedical research including the studies above [20]. At the institutional level, the Charité in Berlin has developed Open Research monitoring with staff incentives, reporting data sharing rates of 7.8% in 2021 using both Open and some FAIR criteria [21]. The methods available to assess data sharing are also changing. The manual scoring required to handle earlier, highly variable sharing practices is now supplemented by programmatic analysis of standards-compliant online infrastructure, with specialised systems to assess the 17 principles of FAIR data [16,22,23].

Bioscience research exemplifies the balance between open accessibility and data protection. Open Access to research articles from our university has been routine since a UK funder mandate in 2014 [24] but this policy did not apply to research data. The University of Edinburgh updated its Research Data Management policy in 2021 [25], to include adherence to FAIR principles and GDPR compliance. Therefore, our research aims to assess the data-sharing practices within the biosciences at the University of Edinburgh from 2014 to 2023. We seek to provide a detailed analysis of how different research areas—biotechnology, regenerative medicine, infectious diseases, and non-communicable diseases—perform

in terms of data sharing. Additionally, we investigate the correlation between Data Availability Statements (DAS) and preprint status on the quality of data sharing. By highlighting the current state of data-sharing practices and identifying areas for improvement, this study aims to contribute to the ongoing efforts to promote transparency and reproducibility in scientific research.

## Methods

### Definition of data

The term "Data" refers to both qualitative and quantitative information necessary to reproduce research results. This includes primary data, such as raw experimental results; secondary data, like derived analytical values, and the software code used to derive those values. Data also includes images and videos used for conducting analyses. When quantitative analysis is performed on images, these images are considered raw data, and the resulting numerical values are considered processed data. Images and videos used solely for illustrative purposes are excluded.

### Sampling framework and selection process

Research groups within the biosciences at the University of Edinburgh were selected for study, based on the educational interests of five student researchers. 57 groups were selected from the University's public research websites in the following areas: Non-Communicable Diseases (NCDs), Infectious Diseases and their treatments (InfD), Microbial Biotechnology (BioTech), Stem Cells and Regeneration (SRM). In each area, all research groups in the School of Biological Sciences were selected. Of the 57 research groups included in the study, 17 were based in the College of Medicine, not in the School of Biological Sciences, but focused on similar research areas. This allowed each student to score and analyse around 100 articles within their project time limit.

Articles were retrieved from the University of Edinburgh's public-facing research portal, Edinburgh Research Explorer (https://www.research.ed.ac.uk/) [26], This portal is powered by the Elsevier Pure system and serves as a curated institutional research information system, designed to ensure the visibility, discoverability, and accessibility of all research outputs affiliated with the University of Edinburgh. This system is required for the University to fulfill our UK funder's Open Access mandate [24]. Article deposit is mandatory (including imports from Scopus, ORCID, PubMed and other sources). Deposits and authors' affiliations to the University are manually curated by support staff. The system's coverage of research articles is very high, because the University's core funding depends upon it.

To be included in our analysis, a group needed a substantial publication record, with at least 10 journal articles published between 2014 and 2023. From each selected group, we randomly chose journal articles for each year within our study period to ensure a representative sample. Our focus was strictly on original research papers that generated new datasets, excluding other publication types such as reviews, editorials, and commentaries. To avoid bias, we did not select articles by visibility or citations, though these could be explored in future work. This selection process was carried out by five biology students, who manually reviewed and verified each article based on shared scoring criteria (detailed below), that were documented on an internal wiki site. The selected articles and scores were compiled on spreadsheets and combined, with each article categorised by research area.

### Scoring criteria

The scoring system was developed based on those established by Roche et al. [16] and the FAIR Data Self-assessment tool by the Australian Research Data Commons [27]. We tailored the scoring criteria, pertinent to our study's objectives. A key adaptation involved refining the scoring scale from five levels, as used by Roche et al., to four. This reduction was motivated by a need to distinctly address file format and metadata quality without conflating them with raw versus processed data considerations, thereby eliminating potential overlap and ambiguities in evaluations.

While Roche et al. [16] sampled from a popular data repository Dryad, where sampling started from archived datasets, this study involved selecting articles while being blinded as to whether any data was shared. When articles reused previously published data, a similar approach to Hamilton et al. [28] was taken. The four criteria – 'Completeness', 'Reusability', 'Accessibility', and 'Licence' – were established to assess independent aspects of data sharing. The Completeness score was based upon the whole article. The criteria for the quality of sharing, Reusability, Accessibility and Licence, were assessed only for the best-shared dataset, in contrast to Roche et al. [16], to avoid confounding these quality criteria with the Completeness of sharing.

**1 Data Completeness:** To evaluate completeness, we assigned scores from 1 to 4, with 4 indicating complete data sharing necessary for reproducing all analyses and results. The highest score did not require sharing the partially processed data at each analysis step, provided that the raw data and sufficient description of data processing steps were available. Scores of 2 and 3 indicated minor and major data omissions, respectively. This ranking accommodated the wide variety of data types and volumes in individual articles and between articles in our sample (see Discussion), as in Roche et al. [16]. When a score of 1 was assigned, indicating no data sharing, the subsequent criteria were also assigned a score of 1 (Not Scored). The absence of shared data negated the possibility of evaluating reusability, accessibility, and licence clarity. We assessed the completeness of data sharing, irrespective of its location – whether in texts, figures, tables, supplementary materials, or external repositories (Table 1).

**2 Data Reusability:** Reusability scores, ranging from 1 to 4, were based on data being in a non-proprietary, human- and machine-readable format, and accompanied by informative metadata. A score of 4 required fulfillment of all these aspects. The criteria for classifying files according to their non-proprietary, human- or machine-readable format are detailed in S1 Table of the S1 File, adapted from Roche et al [16]. The inclusion of metadata in this criterion highlights its indispensable role in preserving the data's long-term utility (Table 1). Proprietary formats were penalised unless they were community-accepted or mandated by public repositories.

**3 Data Accessibility:** Accessibility was assessed on whether the data was accessible, had a unique or persistent identifier (PID) and was archived in a public repository. Scores ranged from 1 to 4, with 4 indicating fulfillment of all conditions. Data with a PID in a public repository but inaccessible or requiring permission received the lowest score, reflecting our focus on data openness. Scoring details are in Table 1.

**4 Data Licence:** The licence type was scored from 1 to 4. Open licences received the highest score, restrictive licences scored 3, and data without a specified licence or governed by the article's licence scored 2. Criteria are detailed in Table 1.

This approach allows for a nuanced assessment of data sharing practices, together with efficient manual scoring. FAIR guidelines did not exist in 2014 when our analysis starts, so specific evaluation of all the FAIR criteria would be most relevant in only the later years of our sample. Numerous datasets lacked sufficient metadata, for example, let alone machine-readable metadata suited to programmatic assessment. Our manual scoring system and the ODDPub software (see below) were better suited to this context and both evaluate articles, compared to more detailed and specific FAIR evaluation of datasets (see Discussion).

### Additional data evaluation criteria

Our study assessed the sharing of three key data types in biosciences research, using a consistent scoring system. Data types—genomic, image, and human—were classified based on explicit mentions in the methods or results sections of each paper. Genomic data included nucleotide-based assays such as transcriptomics as well as genomic sequences but not PCR primer sequences; image data referred to primary data images suitable for reanalysis, such as microscopy; and human data included personally-identifiable information, human cell lines, or participant-derived data. For each relevant data type, we applied a scoring system: 'Not Applicable' (NA) if the data type was not used in the study, 'Not Shared' (0)

**Table 1. Scoring Criteria used for the assessment.**

| Data Completeness | | |
|---|---|---|
| Score | Description | Criteria |
| 4 | Exemplary | All the data necessary to reproduce the analyses and results (in practice) are present within the article (e.g., as supplementary information, figures) or archived in external repositories. Both raw and processed datasets from all methods utilised and mentioned in the study are provided. |
| 3 | Good | Most of the data necessary to reproduce the analyses and results (in practice) are present within the article (e.g., as supplementary information, figures) or are archived in external repositories. Processed data from all methods utilised and mentioned in the article are provided, lacking only a small amount of raw datasets. |
| 2 | Large Omission | Main analyses in the paper cannot be redone because essential datasets are missing AND/OR only summary statistics (e.g., means, standard deviation) obtained from methods utilised and mentioned in the article are archived, no raw data provided. |
| 1 | Poor | Neither processed nor raw data are archived/present in the article OR the incorrect data are archived. |
| Reusability | | |
| Score | Description | Criteria (For best dataset) |
| 4 | Exemplary | Good formats and metadata. Data is archived in a (1) non-proprietary, (2) human- and machine-readable file format that facilitates data aggregation and can be processed with both free and proprietary software (e.g., csv, text); other formats were allowed as *de facto* community standards (see S1 Table, S1 File). (3) Highly informative metadata (such that column headings, abbreviations, and units can be understood in isolation from the original paper). |
| 3 | Good | Data is archived in (1) a non-proprietary OR (2) a human- and machine-readable file format that facilitates data aggregation and can be processed with both free and proprietary software. OR (3) Metadata must at least be sufficiently informative to be understood when combined with the paper. |
| 2 | Average | Poor formats and metadata. Data is archived in (1) a proprietary OR (2) human- but not machine-readable file format (e.g., pdf, jpeg). AND (3) Metadata is not sufficiently informative when combined with the paper. |
| 1 | Poor | Not Scored |
| Accessibility | | |
| Score | Description | Criteria (For best dataset) |
| 4 | Exemplary | Data is accessible, has a Persistent Identifier (PID) assigned or a unique identifier from a high-tier repository (such as a GenBank genInfo number), and is stored in an online public repository (e.g., Figshare). |
| 3 | Good | Data is accessible, has either a PID or a unique identifier from a high-tier repository OR is stored in an online public repository. |
| 2 | Average | Data is accessible but does not have a PID or unique identifier and is not stored in an online public repository (e.g., data shared in the supplementary information section of the article). |
| 1 | Poor | Not Scored |
| Licence | | |
| Score | Description | Criteria (For best dataset) |
| 4 | Exemplary | Data has a permissive licence (e.g., CC0, CC-BY) and code has an Open Software licence (if applicable). |
| 3 | Good | Licence(s) are present but not all are permissive. Data has a restrictive licence (e.g., CC-BY-SA/NC) and/or code has a Closed Software licence (if applicable). |
| 2 | Average | No explicit licence is provided for data or software. |
| 1 | Poor | Not Scored |

if relevant data were generated but not made available, and 'Shared' (1) if the data were both relevant and archived in a reusable format. We also recorded the storage location of shared data, using a separate variable: 'Repository' for external repositories, 'Supplementary Material' if included in the publication's supplementary files, 'Both' for cases with data in both locations, 'DOI in Methods' where the DOI of a previously-shared dataset, re-used in the article, was cited rather than a newly-shared dataset, and 'NA' where no data were shared. Additionally, we tracked the publication of corresponding preprints, using Google Scholar, medRxiv and bioRxiv preprint servers for verification.

Data Availability Statements (DAS) in each article were only recognised if explicitly titled as such or with similar designations. DAS were categorised as 'Shared' when indicating unrestricted data access, or 'Not Shared' when access was conditional. Articles without a specific DAS section were marked as 'Not Presented`. We assume that the inclusion of a DAS reflects journal policy more than researcher preference [20]. Journal policies were not scored directly, because the historic policy versions that applied upon publication back to 2014 were not reliably available. Finally, we assessed the use of analytical software in the research by recording whether it was mentioned ('Yes' or 'No'). Additionally, for analyses that required coding, we evaluated whether the code was shared ('Shared'), used but not shared ('Used and Not Shared'), or not applicable ('NA'), based on the documentation within the publication.

### Data extraction and quality control assessment

The collected data, including both scoring assessments and supplementary variables, were cataloged in an Excel spreadsheet. To test the reliability of our assessment, 15% of the papers from each evaluator's dataset, from a random selection supplemented to cover the entire spectrum of scoring values and years, was re-scored by another researcher. This blind reassessment returned a congruence in scoring assessments between the original and secondary evaluations from 86% to 90%. Consequently, the initial assessments were retained for final analysis.

### Integration of ODDPub algorithm

We applied the Open Data Detection in Publications (ODDPub) text mining algorithm to further evaluate the findings and to explore the automation of open science practice assessment across our dataset. Originally developed for and applied in biomedical datasets, as documented in Riedel et al. [29], ODDPub assists in systematically identifying open data and code practices from full text documents of research articles. We retrieved publication full texts in PDF format from DOIs using the Find Full Text function in EndNote 20 (RRID:SCR_014001) or hand searching. We converted PDF files to text format using the pdftools R package [30] for processing by ODDPub, via the ODDPub R Package. ODDPub provided several key outputs: *is_open_data* is a binary indicator noting whether open data was detected, and *open_data_category* indicates the type of data sharing (e.g., supplement, general-purpose repository, field-specific repository), or records 'NA' if no data is found. Additionally, *is_open_code* denotes whether open code was detected (TRUE/FALSE). Supplementary to these, *open_data_statements* and *open_code_statements* extract text related to their respective sharing details, recording 'NA' if no reference is made to sharing in the text.

To evaluate the performance of ODDPub in our context, we calculated its sensitivity and specificity. Sensitivity, or true positive rate, measures the proportion of actual positives correctly identified by the algorithm, while specificity, or true negative rate, measures the proportion of negatives correctly identified (Supplementary material file 2). These metrics were used to compare ODDPub's automated detection results with our manual assessment. This hybrid approach—integrating manual evaluation with automated scoring—provides a deeper validation of ODDPub's performance and highlights both the strengths and limitations of automated detection methods, particularly when documentation practices vary.

### Statistical analysis

The descriptive statistics were presented as frequency and percentage. An ordinal regression model was used to test for changes over time in the four scoring criteria from 2014 to 2023. Additionally, ordinal regression analysis measured the influence of research area and other sharing variables such DAS, and preprint status on the scoring criteria. To account for potential intra-group similarities in data sharing practices, random effects for research groups were included in the regression models, positing that research groups may exhibit consistent data sharing behaviors. Significance was set at $p < 0.05$, and results were expressed as odds ratios with corresponding 95% confidence intervals. Additionally, thorough checks were conducted to validate the assumptions underlying each statistical model used in this analysis. Details of these assumption checks, along with the complete analytical methodology, have been documented and are available on

Zenodo (https://doi.org/10.5281/zenodo.14169649.). The analyses were performed using R and RStudio Software (Version 4.2.2). The data and code have been shared in accordance with the FAIR principles [6].

## Results

### Article characteristics

From 2014 to 2023, the School of Biological Sciences at the University of Edinburgh published 3706 research papers, as recorded in the Edinburgh Research Explorer [26]. Our study analysed 555 papers, representing approximately 15% of the total publications. This sample was divided into four research areas, with each area comprising between 20% and 35% of the sample (Table 2).

   We also observed varying levels of openness in research practices. Regarding the Data Availability Statement, approximately half of the papers did not present this statement in their publication (see detailed discussion below). Most papers (492/555) identified the programmes used for data analysis. Notably, among the 191 papers that required code for analysis, 112 actively shared their code, indicating a strong tendency towards openness in computational research. However, our findings show limited adoption of preprints, with only 124 out of 555 papers (22%) having associated preprints (Table 2).

### Assessment of openness and FAIRness criteria in research publications

The analysis of data-sharing practices among the four research areas reveals distinct trends. Overall, 86% of the publications shared some or all of their data (Completeness Score >1), with Biotechnology having the highest percentage of papers sharing data (95%). Infectious Diseases followed with 88% sharing data, while Non-Communicable Diseases and Regenerative Medicine had the lowest rates at 79% and 77% respectively (Table 2).

   In the Completeness criterion (Fig 1), only 105/555 of the publications shared all available data, achieving a score of 4. Most articles (66%) received scores of 2 or 3, indicating moderate data sharing, while 14.2% of the publications scored 1, meaning no data were shared. The 79 articles with a Completeness score of 1 were not scored for the other criteria (where 1 indicates Not Scored).

   The quality of data sharing in the remaining 476 articles was assessed by scoring the best-shared data set, which was often shared very well. The largest segments of publications scored the maximum (4) for Reusability and Accessibility,

Table 2. Sharing Practices for Data, Code, Analysis Programs, and Preprints across four bioscience fields.

| | | Bio-Technology (BioTech) (n = 130) | Infectious Diseases (InfD) (n = 198) | Non-Communicable diseases (NCD) (n = 121) | Stem Cells and Regeneration (SRM) (n = 106) | Total (n = 555) |
|---|---|---|---|---|---|---|
| **Preprint** | Yes | 20 (15.38%) | 48 (24.24%) | 31 (25.62%) | 25 (23.58%) | 124 (22.34%) |
| | No | 110 (84.62%) | 150 (75.76%) | 90 (74.38%) | 81 (76.42%) | 431 (77.66%) |
| **DAS** | Open Data | 47 (36.15%) | 75 (37.88%) | 47 (38.84%) | 47 (44.34%) | 216 (38.92%) |
| | Restricted Data | 9 (6.92%) | 21 (10.60%) | 21 (17.36%) | 14 (13.21%) | 65 (11.71%) |
| | Not presented | 74 (56.92%) | 102 (51.52%) | 53 (43.80%) | 45 (42.45%) | 274 (49.37%) |
| **Analysis Program** | Yes | 97 (74.62%) | 183 (92.42%) | 120 (99.17%) | 92 (86.79%) | 492 (88.65%) |
| | No | 33 (25.38%) | 15 (7.58%) | 1 (0.83%) | 14 (13.21%) | 63 (11.35%) |
| **Code Sharing** | Used and Shared | 17 (13.08%) | 24 (12.12%) | 37 (30.58%) | 34 (32.08%) | 112 (20.18%) |
| | Used and not Shared | 3 (2.30%) | 29 (14.65%) | 19(15.70%) | 28 (26.42%) | 79 (14.23%) |
| | Not Used | 110 (84.62%) | 145 (73.23%) | 65 (53.72%) | 44 (41.51%) | 364 (65.59%) |
| **Data-Sharing** | Not Shared (Completeness = 1) | 6 (4.62%) | 23 (11.62%) | 26 (21.49%) | 24 (22.64%) | 79 (14.23%) |
| | Data Shared (Completeness>1) | 124 (95.38%) | 175 (88.38%) | 95 (78.51%) | 82 (77.36%) | 476 (85.77%) |

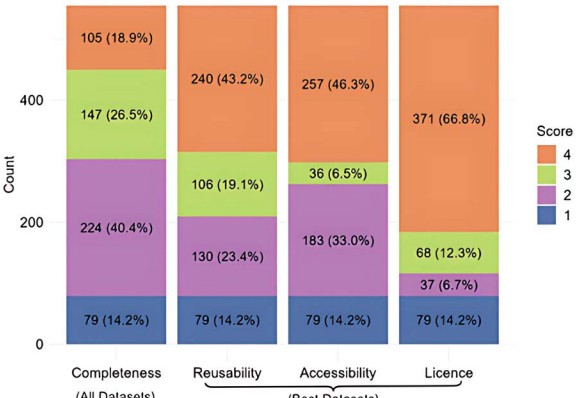

**Fig 1. Scores of the four Criteria (Completeness, Reusability, Accessibility, and Licence) in all the research publications (n = 555).** Completeness is scored for all datasets required to reproduce the article, from 1 (no Open data) to 4 (all data are Open). The other criteria only score the best-shared dataset in each article, to avoid confounding these criteria with Completeness. The fractions of higher scores are at the top of each column, colour code as shown in legend; for scoring details, please see Methods.

with 43% and 46%, respectively. Permissive licensing was even more widespread, with fully 67% of publications having a permissive license for their shared data and (if applicable) an open software license (Fig 1).

### Decadal improvement in sharing, and regression models

Plotting the scores over the past ten years revealed a gradual improvement across all four criteria (Fig 2). The percentage of publications achieving a completeness score of 4 increased significantly, rising from 7% in 2014 to 45% in 2023. Similarly, the other criteria—Reusability, Accessibility, and License—also showed notable improvements over the years, with publications achieving higher scores. These improvements were statistically significant according to the ordinal regression models (S2 Table, S1 File) and remained significant when the research groups were included as a random effect in the regression model.

### Effect of the research area on the regression model

The distribution of scores varies among the four research areas. The articles in Non-Communicable Disease research show a higher frequency of lower completeness scores (S1 Figure), and articles in Regenerative Medicine achieve more top scores in reusability. Ordinal regression analysis (S3 Table, S1 File) confirms that Non-Communicable Disease articles show significantly lower completeness scores, when compared to Biotechnology as the reference research area, indicating substantial gaps in data sharing. Regenerative Medicine papers demonstrate higher reusability scores, suggesting better practices in data formatting and description. The Accessibility scores do not differ significantly across the research areas and Licence scores are high in all areas, so variation between them is not a pressing concern.

Our approach selects articles to study from the publication lists of research groups that focus on each research area, rather than assigning each article to a research area. Patterns that emerged at the research area level might therefore have reflected variations in data sharing among the groups, rather than varying practices across research areas. However, the findings above remain significant when the research groups are included as a random effect in the statistical models (S3 Table, S1 File), consistent with differences in data sharing across research areas.

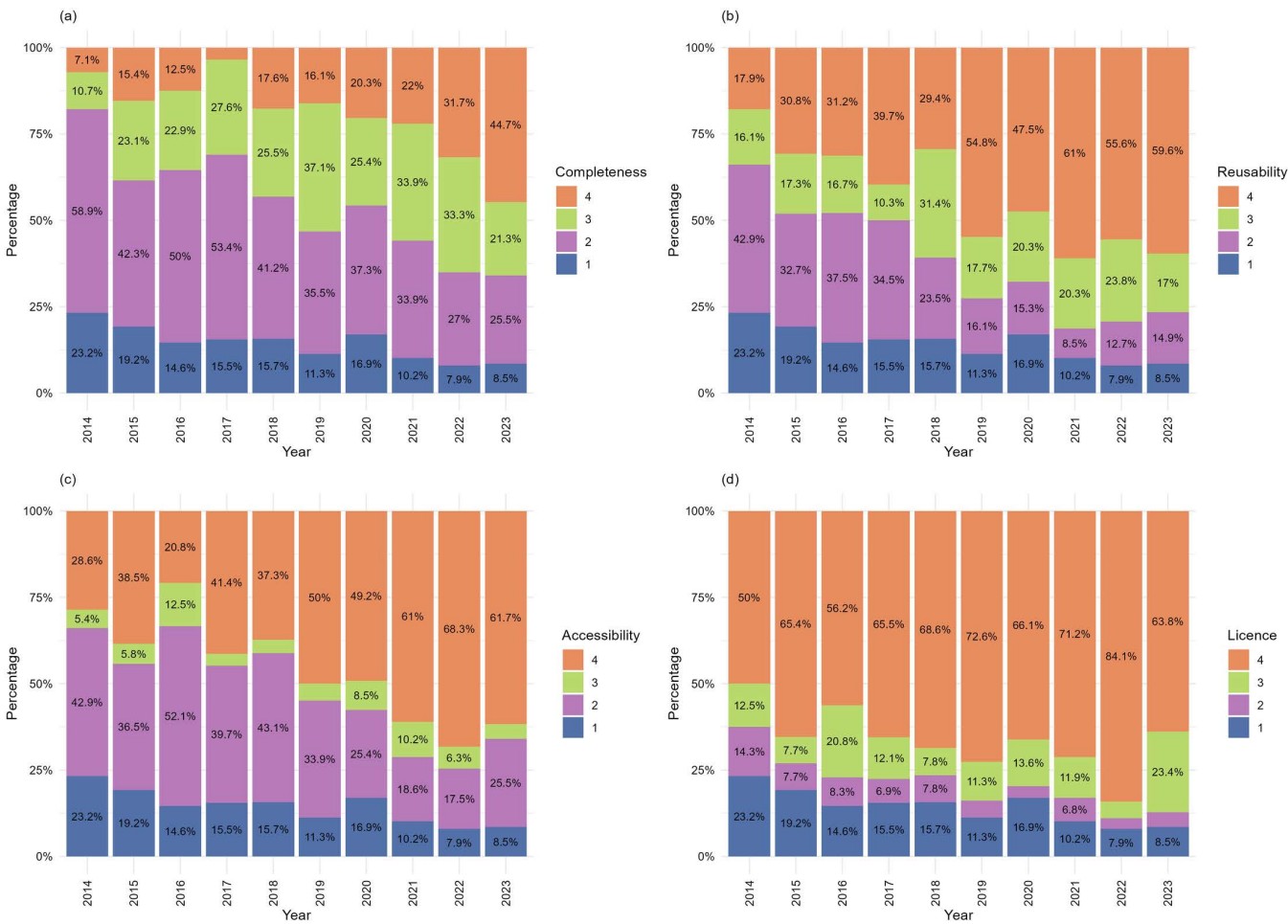

**Fig 2. The distribution of criteria scores over ten years.** Increasing (a) Completeness, (b) Reusability, (c) Accessibility and (d) Licence are shown in the scores of n = 47–62 research publications per year. Fraction of higher scores at the top of each column, colour code as shown in legend, as in Fig 1.

## Comparison of data-sharing practices across research areas

The comparison of data-sharing practices across four research areas reveals distinct patterns in the utilisation and sharing of genomic data. Among the papers that used genomic data, Regenerative Medicine stands out, with 94.3% actively sharing this data. In contrast, only 50% of the Biotechnology papers that utilised genomic data (27 out of 54) shared it. The research areas of Infectious Diseases and Non-Communicable Diseases demonstrated moderate sharing rates, with 74.2% (72 out of 97) and 58.3% (42 out of 72) of their genomic-using papers sharing the data, respectively (S4 Table, S1 File).

For image data, Regenerative Medicine again leads with 29.2% of papers sharing images. Biotechnology, Infectious Diseases, and Non-Communicable Disease papers share image data at lower rates. Human data sharing is less common across all research areas, with Infectious Diseases leading at 14 out of 43 papers. Non-Communicable Disease and Regenerative Medicine papers have a lower rate of sharing human data, while the Biotechnology papers in our sample do not use human data (S4 Table, S1 File).

In terms of data storage, the article's supplementary materials were the most common storage location across all research areas, particularly in Biotechnology (53.1%) and Infectious Disease (37.9%). Regenerative Medicine showed a preference for using both repositories and supplementary materials, with 45.3% (S4 Table, S1 File).

## Effectiveness and Accuracy of ODDPub in detecting Open Data Practices

The effectiveness of the ODDPub text mining algorithm in identifying open data and code sharing practices was evaluated by comparing its results against a manually assessed dataset. ODDPub employs a dictionary approach to identify terms relevant to data and code sharing and assigns publications binary TRUE/FALSE scores for several criteria [29]. Fig 3 illustrates how ODDPub outcomes for open data and open code are distributed compared to manual Completeness and code sharing criteria.

Initially, ODDPub's performance in detecting open data was assessed solely against our 'completeness' criterion, showing a sensitivity of 44% and a specificity of 90%. However, ODDPub scoring of Open data considers multiple factors, so we aligned the algorithm's full definition of open data with our four manual criteria (detailed in S2 File). This improved the

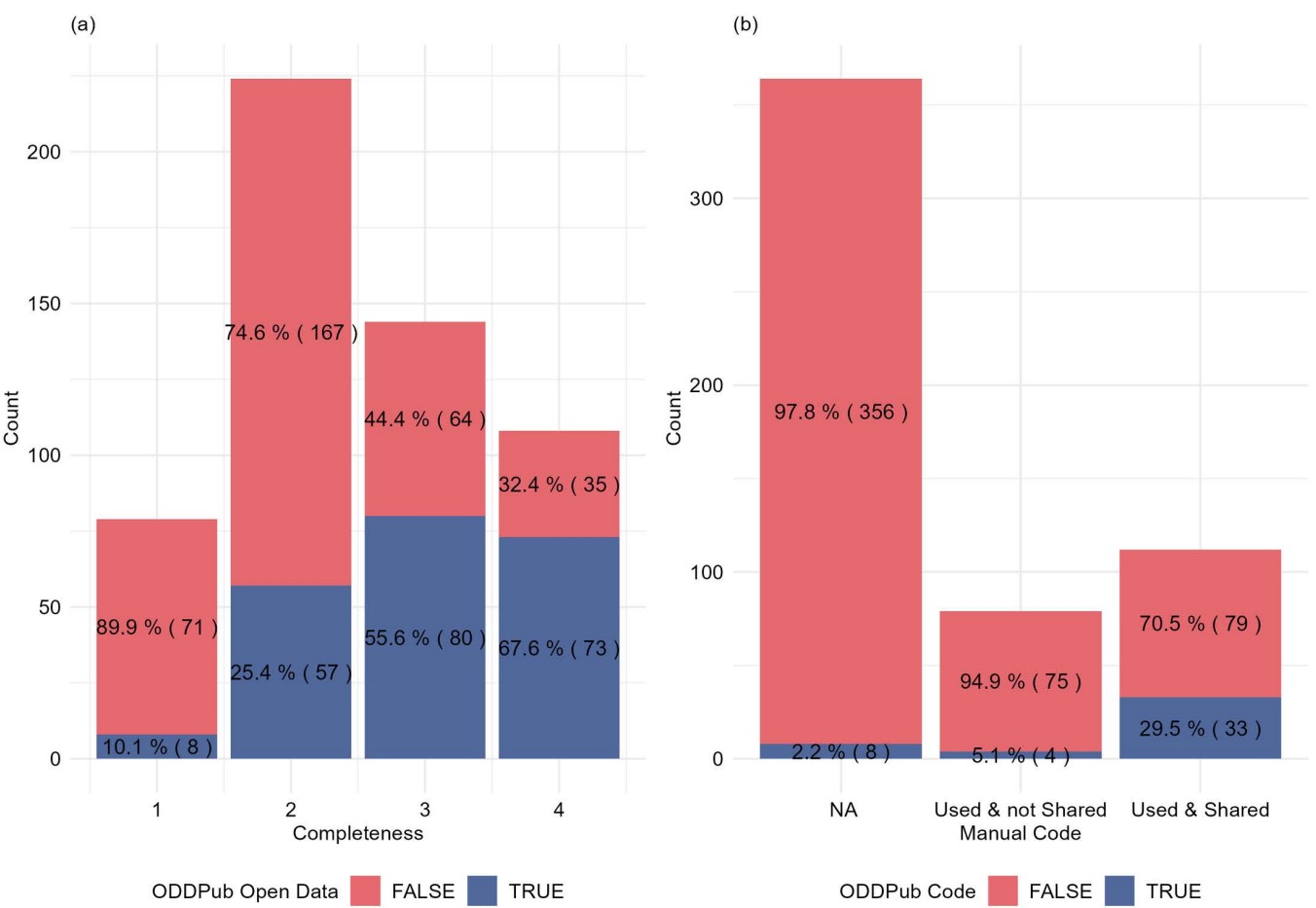

**Fig 3. Manual and automated evaluation of Open Data and Open Code.** (a) Distribution of ODDPub Open Data score (Is_Open_Data, True/False) across manually-scored Completeness in all articles (N = 555). (b) Comparison of Open Code evaluation between the Manual assessment and ODDPub Open Code score (Is_Open_Code, True/False).

sensitivity to 52% while maintaining a high specificity of 90%. These metrics indicate the algorithm's strong capability to identify non-open data instances. However, it shows limited effectiveness in identifying all the actual cases of open data.

In terms of open code practices, the algorithm recorded sensitivity and specificity rates of 30% and 97%, respectively. These results highlight ODDPub's challenges in detecting true positive cases of open code while reaffirming its proficiency in ruling out instances where code is not shared.

## Impact of data availability statement (DAS) and preprint status on scoring criteria

The ordinal regression analysis reveals significant associations between the presence of a Data Availability Statement (DAS) and preprint status on various data-sharing criteria (S5 Table, S1 File). Papers with a 'Shared' DAS showed significantly higher scores across the four criteria. Papers with a DAS indicating data available upon request ('Not Shared') showed a smaller increase in Reusability, Accessibility, and License, but no significant change in Completeness. The presence of a preprint also significantly increased the odds of higher Completeness scores (S5 Table, S1 File), though

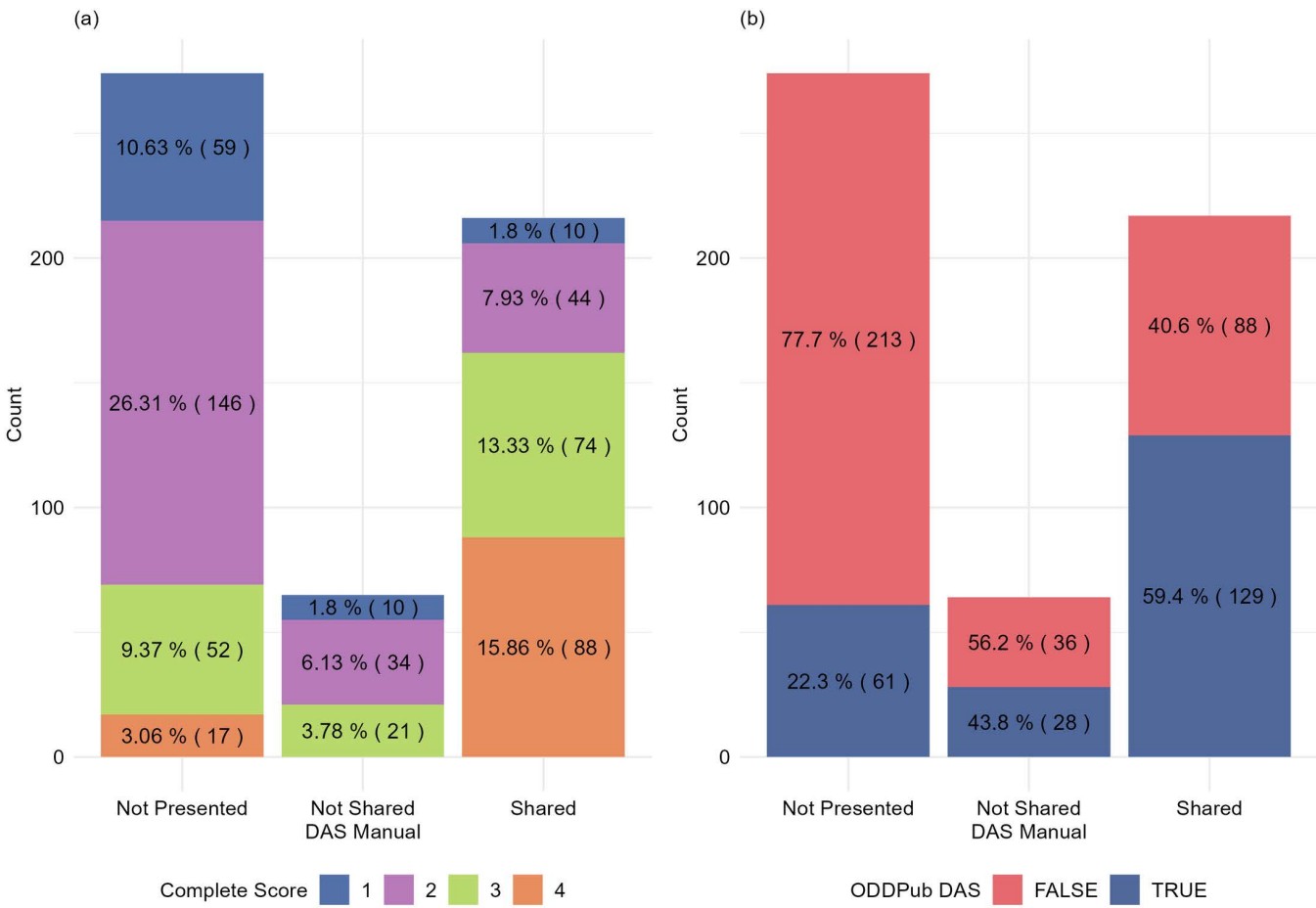

**Fig 4. Manual and automated evaluation of Data Availability Statement (DAS) status.** (a) Distribution of manually-evaluated Completeness scores across manually-evaluated DAS categories for the 555 research publications. Higher completeness scores at the bottom of each column, colour key as in Fig 1. Most articles that shared all data (88 with Completeness 4) included a 'Shared' DAS that signalled this Open sharing. (b) Comparison of Data Availability Statement (DAS) evaluation between the manually-evaluated DAS categories and ODDPub detection, extracted from open_data_category as True/False (see S2 File).

its effect on Reusability, Accessibility, and License clarity was not statistically significant. These findings underscore the positive correlations among data-sharing practices.

**DAS as an indicator of data sharing and its impact on ODDPub detection accuracy**

Fig 4 illustrates the same correlation with Data Availability Statements (DAS), from the converse analysis, stratified by the Completeness score. Papers with higher completeness scores often had a 'Shared' DAS. Specifically, 88 papers with a completeness score of 4 had a 'Shared' DAS, while only 17 papers with a score of 4 had no DAS presented. As completeness scores decreased, the likelihood of lacking a DAS increased. However, 21 articles that indicated data were available upon request (a 'Not Shared' DAS) had in fact shared most of their data openly (Completeness 3), and articles with a 'Shared' DAS were represented in all Completeness categories. At the extreme, ten papers that claimed all data were available in their DAS paradoxically received a Completeness score of 1, indicating that no data were shared. These papers were published between 2014–2021. This discrepancy highlights a gap between the declared availability of data and the actual sharing of complete datasets.

Further analysis reveals that the presence of a DAS, irrespective of its content, considerably enhances the detection capabilities of ODDPub. When a DAS is present, our analysis showed an increase in the sensitivity of ODDPub to 61.8% from the initial 52%, indicating an improved ability to correctly identify instances where data is indeed openly available. However, specificity saw a slight decrease to 85.71% from 90%. This change underscores that while the presence of a DAS generally improves the accuracy of detecting true positives, there may be a minor trade-off in the specificity due to the varied nature of DAS content (S2 File).

## Discussion

Our analysis of 555 bioscience research papers from the University of Edinburgh has shown a progressive shift towards sharing more research data over the last ten years and sharing more "FAIR"ly. Our sample size is larger than most previous studies that assessed institutional research data sharing within medical and health specialties, as discussed in Hamilton et al. [20]. However, our colleagues' research ranges from the atomic scale of molecular structures to populations of intact organisms, so there is no common data unit here, like ecological research [16] and in contrast to the individual participant (or patient) data of many human-centric studies [20].

Our 4-level scoring system evaluates data sharing of journal articles based on Completeness, Accessibility, Reusability, and License, accommodating diverse data sharing approaches. Our Completeness criterion uses our assessors' domain knowledge to assess what datasets were *not* shared, which is challenging to automate. Apart from Completeness, our criteria evaluated the curation and FAIRness of each article's best-shared dataset. The protocol is detailed in a public repository for future reference (see Data Availability) and its broad applicability was recently demonstrated using a smaller dataset in circadian and mental health research [31,32]. Among the possible future developments, our current results could be extended by analysis of article- or journal-level bibliometrics. The scoring approach could be extended to score data volumes, which we discuss below in relation to image data.

To evaluate a more scalable method, we tested the text mining algorithm ODDPub, which also scores data sharing in biomedical journal articles [29]. Our study sample starts in 2014 before the FAIR guidelines existed, making tools that assess specifically FAIR criteria less relevant [22]. Some software tools achieve very detailed and comprehensive assessments of FAIR metrics using machine-readable metadata, starting from a dataset URL or PID. The F-UJI tool [23], for example, assesses 16 out of 17 core FAIR metrics. Our study sample includes many datasets with no URL distinct from the parent article, and little or no metadata, let alone machine-readable metadata, making it challenging to apply such tools effectively. More importantly, approaches that start from a shared dataset cannot identify articles that shared no data, or partial sharing. As ODDPub also scores articles, not datasets, we compared *post hoc* its detection of Open data, Open code and a DAS in the same articles to our manual assessment.

ODDPub identified articles that did not share data with high specificity (90%, Supplemental Material File 2) but its ability to detect shared data was weaker (52% sensitivity), consistent with past reports [21,29]. Our findings suggest that ODDPub is very effective in detecting explicit data-sharing declarations but its sensitivity to nuanced or non-standardised statements may be limited, as expected from its methodology. Where data-sharing rates are low, however, ODDPub analysis can quickly identify at least a significant fraction of the few articles with shared data [33], and its developers have described just such a workflow at institutional level [21].

## Variation among data types

Our manual scoring revealed significant variations in data sharing across data types. Out of 329 papers involving genomic data, 241 shared their data, benefiting from established databases like GenBank and GEO which promote a strong sharing culture among researchers, publishers, and funders [34–36]. This aligns with observations by Kuo et al [37], who noted that the presence of well-established repositories and community norms significantly facilitates data sharing in genomics.

In contrast, only 69 out of 352 papers that used imaging data shared it. General data repositories like Figshare and Zenodo were commonly used, though they lack specialised features for microscopy images and struggle with large data-set uploads (>50GB) due to reliability issues [38]. The BioImage Archive (BIA) only recently started accepting significant contributions [39]. These challenges are further compounded by issues such as the complexity involved in sharing multi-modal imaging data and the high cost of acquiring accurate data labels, as highlighted by Du et al [40]. Due to these challenges, amongst other factors, sharing data in the article's Supplementary Materials also remains a common practice (340 out of 555 articles). These technical and infrastructural challenges extend beyond mere data sharing; they also critically impact data storage during the research phase. Managing large image datasets, in particular, can become expensive and logistically demanding, as highlighted by recent findings [41]. This reality underscores the need for more data management and infrastructural support to facilitate data handling and accessibility [42].

Sharing human data presents additional hurdles, including privacy [42,43], ethical considerations [44], and regulatory challenges [45]. Of 159 papers involving human data, only 48 shared their data, highlighting a need for further exploration of this moderate sharing rate [46], for example to test the motivations and skills among researchers [47]. The fact that many papers in our sample used cell lines reduces privacy concerns compared to data directly from human subjects [28]. Moreover, our Completeness score focuses on Open data sharing, not considering conditional access to data that might still meet FAIR standards.

## Underlying explanatory factors

We did not directly test what motivated researchers to share their data. Survey data suggests that research data sharing was already increasing globally by 2014 [48], when our sample starts. A slightly earlier study, Roche et al. [16], indicated that 56% of ecology datasets in the Dryad repository from 2012–2013 were incomplete (Completeness score <3). Our analysis for 2014 found less sharing at the level of articles, with 77% of articles that shared any data (and could therefore have been tested by Roche et al.'s method) scoring below 3 in completeness. Assessment starting from a shared dataset clearly differs from our assessment of publications. A 2019 study on global cancer research reported that only 19% of the publications tested made some or all of their data publicly available [28]. However, our 2019 sample from diverse parts of biology but including cancer among non-communicable diseases, showed that 53% of papers shared some or all of their data (completeness >2), suggesting higher data sharing. By 2023, we found pervasive data sharing in our bioscience sample, where only 8.5% of articles shared no data (Fig 2). The same manual scoring of an international sample of articles from the same year in circadian neuroscience and mental health found the converse, only 8% shared any data [31,32]. Such large variations in data sharing between research domains will strongly influence institutional-level results, if the distribution of domains (or data types, as noted above) changes either over time within an institution or among

institutions. Comparing institutional data, ODDPub analysis alone detected open data in 14% of articles in 2021 from the biomedical research at Charite, Berlin [21], or 8.8% of Dutch biomedical articles [49], compared to 47% of our sample from that year, or in 10.7% of articles from Linkoping University, Sweden in 2022 [50], compared to 56% of our sample that year. Even in 2014, ODDPub found 20% of our sample shared data, indicating that bioscience articles from the University of Edinburgh have higher data sharing than biomedical research elsewhere.

Several policy changes have influenced data sharing practices, potentially contributing to our findings. The articulation of the FAIR principles in 2016 [6], followed by their progressive adoption, set new standards for data management. The University of Edinburgh updated its 'Research Data Management' policy in 2021 [51], improving upon its 2011 guidelines to strongly advocate for research data sharing in alignment with the FAIR principles. Data-sharing mandates during the COVID-19 pandemic, which impacted the UK from 2020, further accelerated these trends [52,53]. Training programs have been identified as effective methods to foster open research practices and data sharing among researchers [38,54]. Our team provided local, bioscience-specific training for researchers from 2021 [38] that became mandatory for bioscience PhD students from 2022, in addition to broader data management training from the University and growing external provision. Our results reflect these influences, showing a rapid increase in data sharing over the last three years. The highest scores for each of our criteria were achieved between 2021 and 2023. The proportion of articles that fully shared all relevant data nearly doubled, rising from 22% in 2021 to 45% in 2023.

Beyond specific policies on research data management, scientific disciplines [55] and research institutions are increasingly considering Open Research principles in research assessment and researcher evaluation. Initiatives like DORA (Declaration on Research Assessment) [56] advocate diverse research outputs in assessment frameworks, including data sharing, while the Coalition for Advancing Research Assessment (CoARA) supports institutions to implement these policies. Proposals for DORA-compliant research assessment emphasise Open Research [57], but current academic incentives are only beginning to reflect this [58]. Aligned with this broader picture, the University of Edinburgh has signed both DORA and CoARA agreements. The current, University-wide process for academic promotion does not explicitly mention any Open Research practices, so this potential incentive is unlikely to be a major factor in our results. However, a review is already underway and Open Research is one theme in broader work on Research Culture at the University, so future policy changes might reinforce the behavioral shifts revealed in our data.

Research publishers' Open data policies, in contrast, have significantly advanced data sharing, which we measure by the presence of explicit Data Availability Statements (DAS). Our approach, scoring only these explicit statements, contrasts with other studies that assess data availability declarations regardless of their format or location within the article [59]. Standardising these statements will not only improve documentation but also make automated algorithms like ODDPub more accurate, helping researchers to gain recognition for data sharing [21]. However, a DAS alone does not guarantee actual data sharing [60]. In our sample, 10 papers had a DAS stating "all data were available," yet they shared no open data, with the latest published in 2021. Sometimes, authors might confuse the visual display of data, like graphs, with sharing underlying numerical datasets. This continued gap in compliance reflects inadequate enforcement by editors and peer reviewers [61]. The potential for addressing these issues was recognised in a 2023 UK Parliament inquiry [62], highlighting ongoing challenges in both policy development and implementation, including adequate repository support to meet the demands of modern research and policymaking.

## Standpoint

The authors worked either at the School of Biological Sciences, University of Edinburgh during the study, in the Biological Research Data Management team (BioRDM), or for EW, in the Collaborative Approach to Meta Analysis and Review of Animal Experimental Studies (CAMARADES) research group in the College of Medicine and Veterinary Medicine. The BioRDM team grew from the need for data management in our interdisciplinary biology research projects in the Centre for Systems Biology at Edinburgh (later SynthSys, now Centre for Engineering Biology), since 2008. The CAMARADES

group develops systematic reviews of preclinical neuroscience research. AJM also chaired the UKRI-BBSRC Review of Data-Intensive Bioscience (2020).

## Supporting information

**S1 File. The additional figures and tables that support our results.**
(DOCX)

**S2 File. ODDPub: This file provides the methodology, the variables and all the results of using ODDPub on our sample of bioscience articles.**
(DOCX)

## Acknowledgments

We extend our sincere gratitude to Professor Simon N. Wood, Chair of Computational Statistics at the School of Mathematics, and his PhD student, Antoni Sieminski, for their invaluable support and expert consultation provided through University of Edinburgh statistics drop-in clinics. We also wish to thank Professor Malcolm MacLeod for his research supervision for EW. Additionally, we would like to acknowledge the University of Edinburgh Research Information Systems team for their assistance in accessing the information systems that were crucial in identifying our colleagues' publications.

## Author contributions

**Conceptualization:** Andrew J. Millar.

**Data curation:** Haya Deeb.

**Formal analysis:** Haya Deeb, Megan A. M. Kutzer, Emma Wilson.

**Funding acquisition:** Andrew J. Millar.

**Investigation:** Suzanna Creasey, Diego Lucini de Ugarte, George Strevens, Trisha Usman, Hwee Yun Wong.

**Methodology:** Haya Deeb, Trisha Usman, Hwee Yun Wong, Megan A. M. Kutzer, Emma Wilson.

**Project administration:** Andrew J. Millar.

**Resources:** Andrew J. Millar.

**Supervision:** Haya Deeb, Tomasz Zielinski, Andrew J. Millar.

**Validation:** Haya Deeb, Suzanna Creasey, Diego Lucini de Ugarte, George Strevens.

**Visualization:** Haya Deeb.

**Writing – original draft:** Haya Deeb.

**Writing – review & editing:** Haya Deeb, Emma Wilson, Tomasz Zielinski, Andrew J. Millar.

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
