## [Decision Letter · Decision Letter 0]

PONE-D-25-03727The rise of Open Data practices among bioscientists at the University of EdinburghPLOS ONE

Dear Dr. Millar,

Thank you for submitting your manuscript to PLOS ONE. After careful consideration, we feel that it has merit but does not fully meet PLOS ONE’s publication criteria as it currently stands. Therefore, we invite you to submit a revised version of the manuscript that addresses the points raised during the review process.

First of all, I would like to thank the 2 reviewers for their important comments. I have the following additional comments:

The paper deals with the issue of openness and FAIRness of data-sharing practices within biosciences at the University of Edinburgh analyzing 555 research papers, and the results highlight improvements in open data practices between 2014 and 2023.

Although further research in this area is pertinent, this study could be certainly local as presented, being difficult to extrapolate to another context. In this sense, I have missed more references that help to establish comparisons between universities, research areas and countries, both in the introduction and in the discussion section.

As suggested also by reviewers, some parts of the methodology and discussion should be improved.

For instance, regarding the methodology, the authors indicate that they obtain the information from the university's own database, but they do not explain how this database is covered and whether it feeds into other international or national databases. Moreover, they do not explain anything about the quality of the papers or the quality of their journals, in general, the explanation of the sample chosen is scarce.

In the section, Sampling Framework and Selection Process, (line 106) it is not clear if total research groups are 57 or from Medicine there are 17 research groups more.

Regarding the Comparison of Data Sharing Practices Across Research Areas (line 287), What is the basis (is explained in methods?) for classifying/categorizing data as genomics, imaging, human data or data warehousing (table S4)?

Moreover, I missed a better justification of the methods used to evaluate the level of FAIRness of the paper sample. There are different and sophisticated tools to analyze the degree of FAIR, with much more and precise items to evaluate each principle, and maybe it would be interesting to at least give an explanation why the authors chose this one and not others.

Data sharing is a highly current topic in both biomedical research and the broader landscape of science policy. It plays a central role not only in advancing knowledge, but also in shaping research evaluation and the metrics by which scientists are assessed. The authors could also discuss what policies are currently in place to address this and whether this has influenced the increased practice.

Regarding the format, Table 2 needs to be improved as it currently has a poor aesthetic.

As far as I can see, I cannot promise acceptance, but I would reconsider an improved version of the current manuscript.

We look forward to receiving your revised manuscript.

Kind regards,

Rut Lucas-Dominguez, PhD

Academic Editor

PLOS ONE

**Journal requirements:** 1. When submitting your revision, we need you to address these additional requirements. Please ensure that your manuscript meets PLOS ONE's style requirements, including those for file naming. The PLOS ONE style templates can be found at https://journals.plos.org/plosone/s/file?id=wjVg/PLOSOne_formatting_sample_main_body.pdf and https://journals.plos.org/plosone/s/file?id=ba62/PLOSOne_formatting_sample_title_authors_affiliations.pdf 2. Please note that PLOS ONE has specific guidelines on code sharing for submissions in which author-generated code underpins the findings in the manuscript. In these cases, we expect all author-generated code to be made available without restrictions upon publication of the work. Please review our guidelines at https://journals.plos.org/plosone/s/materials-and-software-sharing#loc-sharing-code and ensure that your code is shared in a way that follows best practice and facilitates reproducibility and reuse. 3. We note that the grant information you provided in the ‘Funding Information’ and ‘Financial Disclosure’ sections do not match.  When you resubmit, please ensure that you provide the correct grant numbers for the awards you received for your study in the ‘Funding Information’ section. 4. Thank you for stating the following financial disclosure: This project was funded by UKRI-MRC grant MR/X009726/1. EW is funded by a Simons Initiative for the Developing Brain PhD studentship (SFARI #529085).   Please state what role the funders took in the study.  If the funders had no role, please state: "The funders had no role in study design, data collection and analysis, decision to publish, or preparation of the manuscript." If this statement is not correct you must amend it as needed. Please include this amended Role of Funder statement in your cover letter; we will change the online submission form on your behalf.

Reviewers' comments:

Reviewer's Responses to Questions

**Comments to the Author**

1. Is the manuscript technically sound, and do the data support the conclusions?

Reviewer #1: Yes

Reviewer #2: Yes

2. Has the statistical analysis been performed appropriately and rigorously? 

Reviewer #1: Yes

Reviewer #2: Yes

3. Have the authors made all data underlying the findings in their manuscript fully available?

Reviewer #1: Yes

Reviewer #2: Yes

4. Is the manuscript presented in an intelligible fashion and written in standard English?

Reviewer #1: Yes

Reviewer #2: Yes

5. Review Comments to the Author

**Reviewer #1: ** • The study presents the results of original research.

The research analyzed significantly contributes to the study of open practices in biological sciences through the evaluation of the ODDPub algorithm for assessing open data. Its originality lies in the comprehensive and longitudinal evaluation of open data practices among bioscientists within a specific institutional context, the University of Edinburgh, covering the period from 2014 to 2023. This approach adds novel insights into trends and factors driving changes in scientific data management and sharing practices.

The originality of this work is highlighted by its comprehensive and longitudinal evaluation of the evolution of open data practices among bioscientists at the University of Edinburgh, spanning a significant period (2014-2023). This specific institutional approach provides novel insights into trends and determining factors in scientific data management and sharing practices.

• Results reported have not been published elsewhere.

The results presented in this article are original, contributing novel findings that have not been previously reported in the literature. This work significantly enhances the understanding of open practices in biological sciences, providing new insights into the evolution of open practices.

• Experiments, statistics, and other analyses are performed to a high technical standard and are described in sufficient detail.

The methodology employed demonstrates a high technical standard, clearly defined through criteria of completeness, reusability, accessibility, and licensing. The complementary use of the automated ODDPub algorithm substantially strengthens the study, providing additional important validation by revealing the algorithm's sensitivity and specificity, as well as its strengths and limitations.

This section could be further enriched by incorporating additional references to previous works employing similar tools (e.g., Riedel et al., 2020 or Larsson, 2025) to clearly highlight the advantages of this hybrid methodology (manual and automated).

• Conclusions are presented in an appropriate fashion and are supported by the data.

The conclusions of the study are appropriate and solidly supported by the obtained data. Specific areas such as genomic data, imaging data, and human-related data are highlighted, where significant challenges remain in enhancing openness and quality in data sharing. Concrete recommendations provided could be further strengthened by citing complementary studies addressing similar challenges in other institutional or disciplinary contexts. The authors also offer practical recommendations for improving these practices.

• The article is presented in an intelligible fashion and is written in standard English.

The article is clearly, precisely, and academically written, with a logical and effective structure that facilitates comprehension. Effective use of figures and tables significantly enhances the presentation.

A possible improvement could be the inclusion of specific examples or additional citations about practical guidelines for writing Data Availability Statements (DAS), such as those provided by scientific journals (e.g., PLOS ONE).

• The research meets all applicable standards for the ethics of experimentation and research integrity.

The research fully complies with ethical norms and scientific integrity. It explicitly includes a suitable and transparent "Ethics Statement," declaring no conflicts of interest and clearly detailing funding sources.

• The article adheres to appropriate reporting guidelines and community standards for data availability.

The study strictly adheres to community and editorial guidelines regarding data availability and transparency. All data and codes are publicly available in recognized repositories such as Zenodo and GitHub, ensuring transparency and reproducibility.

This section could be further strengthened by explicitly mentioning adherence to FAIR principles and referencing recent publications or guidelines on these principles (Wilkinson et al., 2016).

It strictly adheres to community and editorial guidelines regarding data availability and transparency, providing complete public access to data and code through recognized repositories such as Zenodo and GitHub.

In summary, this article represents an original, significant, and rigorously documented contribution to the study of open data practices in biosciences, fully meeting all required technical, ethical, and editorial criteria. Its acceptance for publication is recommended, subject to minimal editorial modifications.

It would be beneficial to further explore the reasons behind observed differences in data sharing practices according to data types (genomic, imaging, and human). Including additional references to studies specifically addressing these technical and ethical challenges could further strengthen the discussion and comparative framework of the article (e.g., Kim, 2022 for genomic data; Williams et al., 2017 for imaging data; and Erlich & Narayanan, 2014 for human data).

**Reviewer #2: ** The theoretical framework is appropriate, as it explains the privacy or confidentially concers in a complex field such as bioscience research. However, what is missing is a more precise delimitation of what is the frontier of knowledge in this field, i.e., what is known and what is not known about the subject under analysis, why is it important and what are the benefits of sharing data, concerns, legislation or mandates and challenges? It would also be desirable to have a more detailed description of what are the characteristics and complexity of data in this field, what type of data predominate, in what formats, what difficulties can be encountered in its management and deposition, etc.

In the Discussion section, it is necessary to debate the advantages and disadvantages of the assessment system used by the authors of the manuscript with respect to the use of automated tools already developed and tested, such as F-UJI - Automated FAIR Data Assessment Tool, developed by the FAIRsFAIR project (https://www.f-uji.net/).

They should discuss the importance of awareness training programs in data management for researchers, especially in the field of knowledge addressed.

Discuss whether or not the management, storage and curation of data should be a task entrusted to researchers, technical support staff or information science specialists, as well as the advantages and disadvantages of each of these options.

6. PLOS authors have the option to publish the peer review history of their article (what does this mean? ). If published, this will include your full peer review and any attached files.

**Do you want your identity to be public for this peer review?** For information about this choice, including consent withdrawal, please see our Privacy Policy .

Reviewer #1: No

Reviewer #2: No

---

## [Author Response · Author response to Decision Letter 1]

21 May 2025

We provide a full response to all the editor's and reviewers' comments in the Response to Reviewers document. We substantially revised several sections of the manuscript, so providing a plain-text version here is less helpful than the Response with full quotations.

---

## [Decision Letter · Decision Letter 1]

The rise of Open Data practices among bioscientists at the University of Edinburgh

PONE-D-25-03727R1

Dear Dr. Millar,

We’re pleased to inform you that your manuscript has been judged scientifically suitable for publication and will be formally accepted for publication once it meets all outstanding technical requirements.

Kind regards,

Rut Lucas-Dominguez, PhD

Academic Editor

PLOS ONE

Additional Editor Comments (optional):

All comments have been addressed and the manuscript has been updated. My decision to proceed with the publication is to be accepted.

Reviewers' comments:

Reviewer's Responses to Questions

**Comments to the Author**

1. If the authors have adequately addressed your comments raised in a previous round of review and you feel that this manuscript is now acceptable for publication, you may indicate that here to bypass the “Comments to the Author” section, enter your conflict of interest statement in the “Confidential to Editor” section, and submit your "Accept" recommendation.

Reviewer #1: All comments have been addressed

Reviewer #3: All comments have been addressed

2. Is the manuscript technically sound, and do the data support the conclusions?

Reviewer #1: Yes

Reviewer #3: Yes

3. Has the statistical analysis been performed appropriately and rigorously? 

Reviewer #1: Yes

Reviewer #3: Yes

4. Have the authors made all data underlying the findings in their manuscript fully available?

Reviewer #1: Yes

Reviewer #3: Yes

5. Is the manuscript presented in an intelligible fashion and written in standard English?

Reviewer #1: Yes

Reviewer #3: Yes

6. Review Comments to the Author

Reviewer #1: Nothing to say, all the comments made have been adequately incorporated into the text. It can be published without any new changes.

Reviewer #3: Nothing to say because the authors have adequately addressed my comments raised in the previous review

7. PLOS authors have the option to publish the peer review history of their article (what does this mean? ). If published, this will include your full peer review and any attached files.

**Do you want your identity to be public for this peer review?** For information about this choice, including consent withdrawal, please see our Privacy Policy .

Reviewer #1: No

Reviewer #3: No

---

## [Editor Report · Acceptance letter]

PONE-D-25-03727R1

PLOS ONE

Dear Dr. Millar,

I'm pleased to inform you that your manuscript has been deemed suitable for publication in PLOS ONE. Congratulations! Your manuscript is now being handed over to our production team.

Kind regards,

on behalf of

Prof. Rut Lucas-Dominguez

Academic Editor

PLOS ONE